# COVID-19 and sexual violence against women: A qualitative study about young people and professionals' perspectives in Spain

Esther Castellanos-Torres[1], Belén Sanz-Barbero[2,3], Carmen Vives-Cases[1,2]*, CIBER Program of Violence and Young People team¶

**1** Department of Community Nursing, Preventive Medicine and Public Health and History of Science, Alicante University, Alicante, Spain, **2** Consortium for Biomedical Research in Epidemiology & Public Health (CIBERESP), Madrid, Spain, **3** National School of Public Health, Carlos III Health Institute, Madrid, Spain

¶ The complete membership of the author group can be found in the Acknowledgments.
* carmen.vives@ua.es

## Abstract

There is an increasing awareness of the magnitude of different forms of sexual violence (SV), especially in relation to youth. The COVID-19 pandemic has also had a negative impact on different forms of violence against women. In this study, we aim to analyse SV in the COVID-19 lockdown among young people and SV-related services from the perspective of professionals and young people from different sectors in Spain with responsibilities in attending SV and other forms of violence against women-related. A qualitative content analysis was performed on semi-structured interviews with 23 women and men aged 18 to 24 and 15 professionals working with youth and/or in violence against women or sexual violence related services. The sample was from northern, eastern and central regions of Spain. According to the professionals' experience, the COVID-19 lockdown lessened their ability to work on violence prevention. Both informants perceived that sexual violence had decreased in public spaces whereas it increased in digital ones and noticed the silence surrounding violent situations had deepened. However, they differed regarding its impact on sexual violence within intimate partners, mainly due to the lack of awareness of this problem among young men. In regard to violence against women and sexual violence, our results highlight the need to develop protocols for action and improve resource accessibility in crisis contexts.

## Introduction

Sexual violence (SV) refers to "any sexual act, attempt to obtain a sexual act, unwanted sexual comments or advances, or acts to traffic, or otherwise directed against a person's sexuality using coercion, by any person regardless of their relationship to the victim, in any setting including but not limited to home and work" [1]. Therefore, it includes different kinds of violence that can occur both within and outside marriages or dating relationships.

**Data Availability Statement:** All relevant data are within the paper and its Supporting Information files.

**Funding:** This research was supported by the CIBER of Epidemiology and Public Health of Spain; Grant ESP20PI02. The funders had no role in study design, data collection and analysis, decision to publish, or preparation of the manuscript. URL to the funding organism (CIBER of Epidemiology and Public Health of Spain): https://www.ciberesp.es/

**Competing interests:** The authors have declared that no competing interests exist.

There is an increasing awareness about the magnitude of different forms of SV (between couples or not), especially in relation to young people. It has been estimated that 29% (25%-34%) of women over the age of 14 have been exposed to SV throughout their lives worldwide [2]. In the young and adolescent population, last 12-month SV reaches 24% (21%-28%) in women aged 15–29 years and 26% (23%-30%) in women aged 19–24 years [3]. In Spain, it has been estimated that 9.2% of women who have had a partner have suffered intimate partner sexual violence throughout their lives. This percentage is higher in the 18–24 age group, where victimization reaches 13% [4]. Non-partner sexual violence is also higher in younger women, reaching a prevalence of 10.5% in women aged 16–24, compared to 6.5% in women over 15 years [5]. In addition, young people are exposed to new forms of SV through digital media, such as cyberbullying, sexting, stalking, grooming, shaming and doxing. In turn, pornography, a practice associated with the perpetration and victimization of SV, is also becoming more accessible and normalized among young people [6].

Although the figures are alarming, there is evidence that exposure to SV, when compared to another types of violence against women, decreases the likelihood of women using formal support services and even informal support networks [7]. In the case of young women, support from formal resources may be limited by some barriers, such as trying to solve problems themselves, the lack of knowledge on resources, considering professionals are strangers, trivialization of acts of SV in relationships, among others [8]. Furthermore, a lack of trust towards formal resources, as well as stigma in SV situations influence the victim's behaviour to ask for help [9, 10].

Previous research has already warned that there was an increase in different forms of violence against women at home due to the COVID-19 lockdown, when female victims had to spend more time with their abusers and with movements being limited or even not permitted [11–14]. Some studies also pointed out that partner SV increased during lockdown [15], as well as other kinds of sexual violence among non-partners such as sexual harassment in digital media amongst young people [16]. Despite this dramatic increase, the COVID-19 lockdown and preventive measures have had a negative impact on detecting and managing different forms of violence against women [17–20], and, in particular, SV [15] from related formal services. On the one hand, from a professional perspective, studies have reported coordination difficulties, work overload, lack of means and resources, as well as other issues [21, 22]. On the other hand, women faced different barriers to access these services due to the COVID-19 pandemic prevention measures [23]. There are still few studies that contrast such professional perceptions with those of women [24]. Qualitative studies have also revealed how the COVID-19 preventive measures and the economic crisis hindered and strained intimate relationships among young people, although previous studies focusing on their perspective are also limited [25, 26].

In this study, we aimed to analyse SV in the COVID-19 lockdown among young people and SV-related services from the perspective of professionals and young people from different sectors in Spain with responsibilities in attending SV and other forms of violence against women-related resources. Studying the perceptions and imaginaries of both professionals and young women and men may contribute to the limited research about SV among young people and the COVID-19 impact on SV that include both perspectives [24–26].

## Materials and methods

A qualitative study was conducted based on semi-structured interviews with an intentional sample of professionals and young women and men between 2020 and 2021. The COREQ Checklist for reporting qualitative research was filled out and is attached to this paper as supporting information (S1 Table).

## Participants

The selection criteria for the young people were: 1) age (18–21 years and 22–24 years); 2) sex; 3) level of studies (university or not); 4) place of origin (migrant and national); and 5) geographical distribution according to region. In order to select the professionals, the following characteristics of the resources where they worked were considered: 1) type of resource management (public administration or associations); 2) scope of action (approach to SV and youth services); and 3) geographical distribution according to region. The regions included in the study were: Region of Valencia, Balearic Islands, Catalonia, Region of Madrid, the Basque Country and the Region of Navarre.

After an initial online search, the researchers contacted potential participants who were informed about the objectives of the study. The participation rate was 100%. The criteria to finalize information collection was discourse saturation. Finally, 23 young people and 15 professionals were interviewed. The main sociodemographic characteristics of professionals and young people are shown in Tables 1 and 2.

## Data collection

The interviews were conducted by two interviewers with training and experience in qualitative research. Sex criteria were taken into account for conducting interviews with young people: a woman interviewed girls and a man interviewed boys due to the age of the population and the sensitivity of the topic addressed. Therefore, the aim was to guarantee rapport and the establishment of trust to achieve the fluidity of the discourse. The development of interviews with the professionals was not guided by these criteria.

**Table 1. Distribution of interviews with young people according to age, studies, sex and region.**

| Code | Sex | Age | Studies | Origin | Region |
|------|-----|-----|---------|--------|--------|
| Woman_01 | Woman | 23 | Univ. | National | Madrid |
| Man_02 | Man | 24 | Not univ. | Migrant | Madrid |
| Man_03 | Man | 23 | Not univ. | National | Navarre |
| Man_04 | Man | 19 | Univ. | National | Madrid |
| Man_05 | Man | 18 | Univ. | National | Basque Country |
| Woman_06 | Woman | 19 | Univ. | National | Region of Valencia |
| Man_07 | Man | 18 | Univ. | National | Navarre |
| Man_08 | Man | 19 | Univ. | National | Region of Valencia |
| Man_09 | Man | 19 | Univ. | National | Basque Country |
| Woman_10 | Woman | 23 | Univ. | National | Region of Valencia |
| Woman_11 | Woman | 20 | Univ. | National | Catalonia |
| Woman_12 | Woman | 22 | Not univ. | Migrant | Basque Country |
| Woman_13 | Woman | 19 | Not univ. | Migrant | Balearic Islands |
| Man_14 | Man | 24 | Not univ. | Migrant | Catalonia |
| Man_15 | Man | 23 | Not univ. | National | Region of Valencia |
| Woman_16 | Woman | 23 | Not univ. | Migrant | Balearic Islands |
| Woman_17 | Woman | 23 | Not univ. | Migrant | Catalonia |
| Man_18 | Man | 24 | Univ. | National | Catalonia |
| Woman_19 | Woman | 19 | Not univ. | National | Navarre |
| Man_20 | Man | 25 | Not univ. | Migrant | Madrid |
| Man_21 | Man | 18 | Not univ. | National | Balearic Islands |
| Woman_22 | Woman | 19 | Not univ. | National | Balearic Islands |
| Woman_23 | Woman | 24 | Not univ. | Migrant | Basque Country |

**Table 2. Distribution of interviews with stakeholders according to type, field of action and region.**

| Code | Sex | Type | Field | Region |
|---|---|---|---|---|
| PROF31 | Woman | Government | Sexual violence | Madrid |
| PROF32 | Woman | Government | Sexual violence | Madrid |
| PROF33 | Woman | Community | Young people and sexual violence | Balearic Islands |
| PROF34 | Woman | Community | Sexual violence | Catalonia |
| PROF35 | Man | Community | Young people and sexual violence | Madrid |
| PROF36 | Man | Community | Young people and sexual violence | Madrid |
| PROF37 | Woman | Government | Sexual violence | Region of Valencia |
| PROF38 | Woman | Community | Sexual violence | Catalonia |
| PROF39 | Woman | Government | Sexual violence | Navarre |
| PROF40 | Woman | Government | Young people | Basque Country |
| PROF41 | Man | Community | Young people | Basque Country |
| PROF42 | Woman | Community | Young people | Catalonia |
| PROF43 | Woman | Government | Young people | Region of Valencia |
| PROF44 | Woman | Community | Sexual violence | Region of Valencia |
| PROF45 | Woman | Government | Young people | Balearic Islands |

Two thematic scripts were adapted, one aimed at the young population and the other at professionals. The scripts were tested and adapted in the early stages of the fieldwork. The final structure of the script aimed at professionals focused on the perceptions of the informants on the following dimensions: 1) description and assessment of the resource where they work; 2) perception on sexual violence and young people's victimization; 3) young people's difficulties and facilitators in accessing and using resources; and 4) the impact perceived of SV throughout the COVID-19 pandemic among young people and their own services and work. The final structure of the script aimed at young people focused on the perceptions of the informants on the following dimensions: 1) perceptions on sexual violence; 2) experience with this problem; 3) knowledge and assessment (self-perceived difficulties and facilitators) of the existing resources; and 4) the impact perceived of SV throughout the COVID-19 pandemic among young people and the existing SV-related services. This study is focused on describing and comparing the professionals' and young women and men's responses to the fourth topic of both interview guides.

Interviews were conducted by phone (26) or video call (12) due to the COVID-19 pandemic itself, making it difficult to hold in-person meetings and generate interpersonal interactions. The interview format was subject to the possibilities and preferences of the interviewees. The interviews were conducted in Spanish. Different joint work sessions were carried out between the team to reflect on the possible verbal and non-verbal communication strategies to be established to minimize the barriers associated with remote interviews. They lasted approximately 50 to 70 minutes, were digitally recorded and transcribed verbatim.

## Ethical considerations

A protocol was implemented to guarantee the ethical principles of the research and the General Data Protection Regulation (GDPR). The study was conducted in accordance with the Declaration of Helsinki and approved by the Ethical Committee of the University of Alicante (UA-2020-07-07). Due to the COVID-19 mobility restrictions, interviews were conducted by phone or video call, and therefore, written consent could not be obtained. The Institutional Review Board approved use of verbal consent. Documentation on the study was provided and informed consent was given prior to the interviews. In this sense, a paragraph was read at the

beginning of the interviews to inform the subjects about the project and the conditions of participating in it, as well as requesting verbal consent from them.

## Data analysis

A qualitative content analysis was carried out by two of the authors, ECT and CVC (both sociologists and experts in gender violence, sexual violence and qualitative research), using a predefined tree of codes that were agreed upon by all authors (experts in gender violence and sexual violence) and based on the study objectives and a preliminary analysis of information from the interviews. After using this initial tree of codes, new codes that emerged were added during the analysis process. The tree of codes was adapted during the coding process, by unifying codes according to their discursive similarity and divergence with each informant profile; the emerging topics that were constructed explained the elements that impacted SV during lockdown, considering the context from which each informant spoke (professionals and young people). The following measures were used: double coding of interviews, contrasting the usefulness of the emerging codes with CVC and BSB in the analyst and debate sessions with the rest of the members of the research team to clarify discrepancies between the main analysts (the coding tree is shown as a S2 Table). The analysis was carried out manually.

## Results

Two main themes have been identified throughout this study, first of all the impact COVID-19 on professional answers; and second of all, the impact regarding SV. The first one was mainly referred by the professionals, while the second one referred to both the professionals and young people's perceptions. The first theme has been categorized into two aspects related to the negative and positive effects of the COVID-19 preventive measures. These two aspects were described in the following sections: 1) from paralyzing direct activity due to lockdown to searching for alternatives for support and intervention in SV; 2) opportunities of online services: the positive aspects of online interventions; and 3) victim-related challenges. The second theme includes agreements and disagreements among professionals and young women and men due to their knowledge or misunderstandings about SV, intimate partner sexual violence, other types of SV, its visibility (or invisibility) and the settings where cases occurred during and after the COVID-19 lockdown. The following categories are described in the results: 1) SV inside and outside the family setting; 2) an increase in SV in digital settings; and 3) lockdown and a decrease in SV in public; and 4) SV cases silenced during the COVID-19 lockdown.

### The impact of professional answers

**From paralyzing direct activity due to lockdown to searching for alternatives for support and intervention in SV.**   According to the interviewed people who carry out community intervention work, whether at education centres or organizing awareness workshops with young people, lockdown and social distancing due to the COVID-19 preventive measures have decreased the scope of their actions, therefore resulting in a complete halt of some programmes.

> *Since the pandemic began I haven't gone back to working with young people, so imagine, since March (. . .) so the opportunity was lost to do more things in the rest of the course (. . .), this project I have with young people, others were directly cancelled (PRO33).*

> *When lockdown started, the boys and girls stopped going to school and they didn't have a centre of reference (. . .) Now with COVID, everything's gone to shit (PROF35).*

However, due to this parenthesis in face-to-face intervention with young people, there are some who say, as is the case of a leisure and free time monitor or a technician from a sexual diversity entity, that the virtual space has become a place to continue to work, although with certain limitations due to the lack of face-to-face contact and the complexity of SV cases.

*Now we can't do any activities, of any kind, unless they're online. And when we've started to do activities, there's been less people attending (PROF43).*

*Because, of course, we don't have meetings, well we can't talk about these things, because these things take place at workshops and they come to meetings (PROF35).*

Thus, for example, the measures at education centres in terms of COVID-19 restrict face-to-face meetings. This fact contrasts the model of social intervention with young people who require physical contact, so they can express their emotions and feelings, or face-to-face work dynamics in groups, such as role playing. The social distancing measures entail, according to the interviewed professionals' opinion, a handicap regarding the effectiveness of the actions with young people to prevent SV.

*The strong part of all of this is being with your colleagues, sharing things there and then, so contact is really present in our workshops (. . .) online sessions, well we saw that we couldn't continue like this for long, but they wanted to, so they wanted the in-person workshop, of course, it's different (. . .), in education centres, well we can't go for now because, well, because of all the safety measures in place, our workshops entail, to start with, changing the whole classroom. We go to a centre and we remove tables, chairs, we move the classroom around, and we can't do that right now, for example. Well, many activities involve contact or a debate group in small groups to talk about romantic love, violence, pornography (. . .) the workshops won't be like before. To start with, we'll have to wear masks, keep a distance. Well, adapt it, and it won't be the same (. . .), that strength won't be there (PRO33).*

*It's all lost, it's all lost . . . (. . .) several times a student would come and tell me things, right? That doesn't happen on Zoom (. . .) that doesn't happen, of course, it's all lost, there's a distance. There are certain things related to being close to someone, have a bond with a person of reference or from your surroundings (. . .) be part of their everyday life, we bond with people, they tell me things. On the Internet they don't (PROF45).*

Furthermore, professionals who provide direct support to women who have suffered SV coincide in the fact that adapting to this new context has had an impact on the ways to intervene and address cases, especially when going from face-to-face work that generates spaces of trust to reorganizing the way to work with women online to continue providing continuity and following up cases. They considered this new form of addressing victims to be more demanding and stressful.

*This has also marked a before and after (. . .) it was restructured like this and then, we had to do a lot of telephone service, a lot, a lot, a lot. Of all kinds, from first visits, from first visits, that we didn't know them, from people who were already following up, your cases, the cases of the colleagues, emergencies . . . it was, it's really been, an overflow, overflow and at the telephone level (. . .) I've even continued through email (. . .) so that they could write to me, to feel that continuity that we didn't have (PROF37).*

**Opportunities of online services: The positive aspects of online interventions.** In the process of following up the cases of women exposed to violence, this professional expresses that working online can be an advantage in relation to women going in person, precisely because of the very psychosocial consequences that SV has on women and the recovery process itself, as it involves a safe place, when they live alone or have a space of privacy or intimacy. However, it can be a barrier if the woman still lives with the family and cannot travel.

*With videoconferencing, too, which has improved a lot (. . .) as there's a lot of avoidance behaviour, coming here already means they've lost sleep days before . . . because they know that they're going to see me, that is, discuss related events with aggression, with sexual violence. So, they already had, they're a little dissociated, right?, As a coping mechanism and sometimes they feel bad. (. . .) Even if it's by video call, much better than by phone. They do it at home, it's a safe place and from that safe place, they kind of express themselves better. As long as they live alone or have that space, that privacy, okay? If they live with the family, well no, it's another handicap, right? (PROF37).*

On the other hand, this professional believes this type of technology to be advantageous for the interventions with the young population given the familiarity that they have with new technologies and reveals the generational gender digital gap of older women.

*Young people have adapted best to this. Take into account that going to therapy on a platform, for a woman maybe of 60 years of age, it isn't practical. Because maybe she doesn't have a computer, or she has a computer, well her daughter or granddaughter use it and she can't access it. However, young people have got on well, the opposite. They've got on psychologically, with those platforms and so, they've carried on with therapy without any problems. So, in that sense, I think that we have worked well with young people. Or a workshop we've done as well online, no problems. It's what they do on a daily basis, it isn't difficult for them (PROF44).*

**Victim-related challenges.** Those who provide direct service to women, who use specific resources for SV, this context has favoured in some cases telephone consultations. The interviewed professionals express that from their professional experience, SV cases with partners increased during and after the COVID-19 lockdown.

*Yes, there's been an increase in the number of cases, right? After lockdown, lots of new cases started to come in (PROF39).*

*So, now with this de-escalation that we're going through, it's true that we're receiving, well, more cases with young women who've suffered sexual violence (PRO37).*

In addition, they perceive an increase in telephone enquiries from women in different social vulnerability situations, such as unemployment, job insecurity or belonging to an ethnic minority which have made SV perpetration possible.

*We've started to receive a lot, you know? Calls from many women, women already vulnerable, young people without papers, (. . .) who go to find a job, which aren't real jobs and they've been sexually assaulted (PROF34).*

*In the pandemic we've seen the situation within sexual violence, the issues of work, racialized women in highly precarious spaces (. . .) they were a lot lonelier and violence and harassment have also increased (PROF38).*

## The impact perceived by professionals and young people

**SV inside and outside the family setting.**   Lockdown has affected the intensity of the already existing SV that occurred in the family and close surroundings. The general perception of professionals is that SV has continued to occur mainly in environments close to women, living with abusive partners, friends or their close circle. Specifically, the interviewees speak from their professional praxis, not being so much an idea that could be in the social imaginary but rather from their own daily experience at work.

*I can't tell you how many cases we've had this year, but there have been cases still that maybe you think that because we've been confined there hasn't been sexual violence, but there has (. . .) sexual violence still exists, it hasn't stopped during this time (PROF 31).*

*It can't be easy to live for 3 months with someone, well, that's violent towards you. And that scares me (. . .) that they've had to live with their abusers for those months and they've also had that difficulty and they still find it difficult to get out of there (PROF42).*

This opinion expressed by the professionals from their experience and knowledge is in line with young women's stories. However, the young women do so from what they imagine lockdown has meant in cases of SV in closer coexistence environments (couples, family members). This young woman discusses her experience:

*I've seen cases, of girls, well, young girls who have been on lockdown with their partners and everything that it entailed. So, I'm talking about relationships,, well, let's say not the healthiest, right? Then, I have, I've seen it, well, daily threats, coercion, even, what I said, non-consensual relations for the 3 or 4 months we were on lockdown. So, I think that, yeah, it obviously went up (Woman_19).*

However, the young women do so from what they imagine lockdown has meant in cases of SV in closer coexistence environments (couples, family members).

*the lockdown, to me that seems to me to be an absolute madness. That there's been women who have been . . . uff, I can't even think about it. Because the sexual violence must have been horrible (. . .) how can you be 24 hours, 7 days a week, for 3 months, locked up with the same guy, who does the same thing to you every day, or every day something awful, I don't know what's worse (Woman_01).*

*I think women who've been confined with their sexual abuser have gone through hell, complete hell. I can't say so from experience, but I can imagine it. I just don't want to even imagine it (Woman_22).*

In this sense of what is imagined, some of the opinions of young men agree with young women, when some express that violence could indeed occur in lockdown where there had previously been abuse.

*In more stable relationships or living with their partner or abuser, well the fact that living with them all the time and being locked in with that partner has been hell (Man_04).*

*Women had to be locked in with people who abused them in relationships they were in (. . .) but there have been cases of young people who have said: "I'll go through lockdown with my boyfriend, although we're 18 years old, we'll go to my university flat" (Man_05).*

However, they recreate their discourse not only based on what they imagine or project, but their nuances are different, as they do not know of any close cases, they do not name them and they justify them with the anxiety and stress caused by the situation.

*An abuser already with problems that comes from childhood, from his bad education and we add the COVID situation and the psychological pressure that's being experienced, well I'd think that it's likely that sexual crimes . . . and even after . . . or at this time, increase, I don't know, I'm not specialized in psychology, but I think it could affect, it could . . . I don't know, generate more anxiety, more aggressiveness and consequently (Man_20).*

*We young people also have a problem, of course, but how about . . . I mean, there is a machismo discourse, right?, (. . .) the man is biologically stronger and like he continually generates sperm. His instinct asks him to fuck more, right? So, it's like . . . what I'm telling you is crazy, but it's like . . . then, it's justified that men have that dominance and that need more than the women (Man_18).*

**An increase in SV in digital settings.** Another discursive idea is the perception that SV has increased in virtual contexts such as sexting, sextortion or digital sexual harassment. Professional and young women's opinions confirm the current problem of young people being addicted to new technologies, but also the problem of SV and its link to using online pornography.

*One of the consequences that we've seen after lockdown, they've doubled their addiction to new technologies. In other words, we have kids who admit to us that they spend 8 or 10 hours connected (PROF36).*

*This lockdown situation, where I guess more pornography will have been used, more . . .also videogames, right? With that kind of hyper sexualization of women, objectification . . . (PROF37).*

*Sexting has started to be used more and that. Also, incitement and harassment and so, well yes, a lot and during the pandemic as well, I've seen that more in girls, harassment by ex-partners that were maybe bored during the pandemic and have harassed them, well also guys who follow you on Instagram (Woman_19).*

Only one young man agrees with the women.

*Perhaps sexual crimes have increased a bit, but I think they wouldn't be the same without lockdown. And, above all, the. . . and more, above all, the crimes on social networks. I think also even more (Man_20).*

This young woman indicates the importance of a message that was made about SV on social networks and how they can be used as a means to announce an increase in post-lockdown SV and generate threats through these types of messages.

*A video went viral of a guy on Instagram that said: "Girls be ready because when the pandemic ends, guys are going to out and are going to start raping like mad, because they haven't had sex for two months, so we should hide, to not go out anymore". And it got really, really, really famous and I think in the end the guy's account got cancelled (Woman_06).*

This young woman speaks from her experience and how men are placed in these asymmetrical power relations around their sexuality.

*I think that any app where you have to flirt, you'll have a bad experience, without physical violence or with it (. . .), it's happened to me personally and to close friends, too. And . . . well, a bit like . . . there's no way . . . when you don't know . . . With the pandemic [laughs] and so, of course, the way we relate has changed a lot. And there are guys who think that by making a video call with you, you're going to have . . . Well, you're going to have . . . that you're going to end up masturbating with him on camera, come on. And, of course, they start, well: "no, no, we're having a good time, it's a laugh", and so on. But they don't do anything, they just want you to do it, that the only exposed person is you, basically. That you're sending him content, so that that person gets an erection and that's it. Because if it were a give and take, I don't think it would be a problem, because both people are doing it. Well, let's see, it still carries on without the pandemic, right? But I think that a lot happened in lockdown (Woman_23).*

**Lockdown and a decrease in SV in public.** Some young men believe that SV among the young population has decreased thanks to lockdown, as they still live in the family unit. In these cases, they refer to SV outside relationships. In this sense, they believe that SV only occurs in public spaces, in the most common form known as rape.

*In fact, maybe we don't relate so much, there isn't so much leisure, maybe not. (. . .) I don't know. But I think that young people . . . I don't know the data, nor what has happened, but, as well as a hypothesis, I think that maybe it's decreased and the fact of that, of not relating with others so much, not going out as much, going out with your circle of friends, don't go out too much due to the risk of contagion, for whatever reason. Well, yes, maybe the figures have decreased a bit, I don't know (Man_04).*

*We couldn't go out, so, you could only go out for essential things, so I think cases of gender violence and harassment and so, I think they've gone down, well, I'm not informed, it's what I think, so, I think they could've gone down (Man_07).*

However, some professionals agreed that this consideration exists among the young population and shows how SV in a close setting is not so present in the social imaginary.

*We've found situations of sexual violence in a close setting and that's what usually happens, right? (. . .) The myth of the unknown person is false. Well, it's a myth. That the attacker is a stranger late at night returning from a party is a myth (PROF31).*

This perception regarding a decrease in SV also emerges among young men and women. They attribute this to a decrease in SV cases in public spaces (street, leisure places) due to the

restrictions to go out. On the one hand, this is justified by night-time leisure activities being closed and this being a context where sexual assaults can occur, where people drink alcohol and other substances are taken.

> *I think it happened more often before, in clubs with alcohol, with, shit, they're all here, I'll manipulate her, I'll put whatever in her drink, you know? Like closer to young women, they could have . . . (Man_18).*

> *I wouldn't know, I'd say these cases on the street, that happen in an alleyway. I'd say they've gone down, but of course, we couldn't go out, so they went down (Man_21).*

Women see it as a positive impact, as they are the ones who can be raped or sexually assaulted.

> *I think that when you go out to a party, you have that fear of walking the streets and you can be assaulted, I don't think it's happened as much because in the end, you can't go outside much. So, I think that, in theory, in that sense, it's gone down (MJ12).*

> *. . .I think that the majority of sexual assaults usually happen with young people, with unknown people or on the street. Maybe one day you go out . . .one night you go to a club and you drink a lot. I don't know, something is put in your drink, they take advantage of you. So, this lockdown has prevented many of those situations, it happened a lot, you know? (MJ22).*

Young men again justify this stereotypical place of male drives as the cause of SV. Thus, they express that after lockdown SV can increase.

> *When this is over I think SV will be even worse. Everything will be crazy, because people, I don't know, people will be like, as they say, wanting to party (HJ09).*

> *Like when you have a pit bull locked up (. . .) I work with lots of young people, I hear it and they say: "Wow, have you seen how fit she is after lockdown". They've come out of lockdown with lots of hormones (HJ02).*

**SV cases silenced during the COVID-19 lockdown.** The young women interviewed perceive that isolation caused by the pandemic has had even more influence on silencing SV in close surroundings. In these cases, intervention and searching for help, whether formally by reporting to the police or informally by talking to friends, has been negatively influenced by the direct control by their abusers. In the young people's discourse, this idea does not even appear. They feel questioned and express it in the following way:

> *It's much more difficult to get out of there, or to communicate it at the end, because you can't do something as simple as being alone with your friend at the bar, wherever, and then tell them. No longer resorting to help as such, but the fact of being able to tell it in a way that you feel safe, right? Not within the 4 walls with, with the person who may be causing you, who has caused the violence (Woman_11).*

> *I think that any young person who has been suffering from sexual violence by a family member with whom they were living, has been locked up with their abuser and without the possibility of going out and without the possibility of seeking help, so, yes, it seems to me that it is something really awful (Woman_16).*

Only one of the professionals from the university setting imagines and indicates that isolation has resulted in silence and that it can influence revealing SV. This informant highlights one of the measures that was put in place in Spain so that women who were in a situation of risk or danger regarding their physical, psychological and/or sexual integrity, whether within their family or on the street, could go to the pharmacy and ask for a "mask 19".

*When someone suffers some sort of violence, sometimes they're embarrassed. So, when we're isolating, because of the pandemic and with more independent bubbles, well of course, there's more silence. But there's certain campaigns, now in the middle of the pandemic there was a message that, if you were being abused or suffering violence, you could ask for a purple mask at the pharmacy (PROF45).*

## Discussion

This study shows how the COVID-19 pandemic and the adopted public health measures have resulted in the need to adapt support services for violence against women, in general, and SV, in particular. Said service reorganization has been followed by an overload of work for professionals who have had to adapt to working online, which has generated changes in the way they intervene with young people. This study also reveals the perceptions that both professionals and young people have regarding what has happened and what could have happened with a problem like SV in lockdown. Both groups of informants agreed that this situation has worsened SV when abuse and violence already existed, whether within the family or relationship. Furthermore, regarding this situation, lockdown has resulted in a barrier to report situations or ask for help. They also perceived that SV has decreased in public and increased in digital spaces during lockdown.

The results of our study are in line with other studies conducted throughout that period. First of all, those related to adapted services agree with Esther Bennet et al. [16] when explaining that, although many workshops and courses have been cancelled and/or rescheduled, virtual alternatives have continued and been offered. Nonetheless, adapting support, in this context, has been a challenge as mentioned by Johnson et al. [15], indicating that adapting support protocols has caused tension and stress among staff. At the same time, these services tend to have a shortage of base personnel and in crisis situations, when new ways of working must be adapted, there is already a lack of personnel, and the workforce is not reinforced with new people. Therefore, technology can be an ally in order to continue offering services, but in areas such as health, staff have seen that using masks has changed the dynamics in the relationship when offering the service in terms of establishing good communication and expressing empathy. These are keys in the approach to SV. This finding is also in line with our study.

In second place, we have seen a generalized perception especially among professionals regarding the fact that SV in couples has increased and worsened during the COVID-19 lockdown. This finding is in line with other studies that show how the COVID-19 lockdown increases levels of stress and tensions when there was already abuse and violence against women [15, 16, 27, 28]. Family violence increases after large-scale disasters and at times when families spend more time together, such as holidays [29]. In these contexts, power dynamics occur that can be distorted and subverted by abusers, often without the scrutiny of anyone outside the couple or family unit. So staying at home has had important consequences, both for those who already had a violent family context (physical, psychological and sexual abuse), or for those who live with an abusive or controlling father or partner [29].

Regarding silencing SV cases, young women may have had difficulties in seeking formal help due to lockdown and control by their abusers. As mentioned by Janse van Rensburg & Smith [30], the possibilities of intervention decrease when women are isolated at home as communication with support networks (friends and family) and online services are more difficult to conduct. In this sense, if these women were fired from their jobs, furloughed, or had to return home from student accommodation, these authors suggest that these situations reduce women's connections with other people and the opportunities to leave abusive homes. Furthermore, physical distancing measures reduce the ability for others to observe or ask questions regarding signs of abuse. Despite the fact that care services for gender-based violence were considered priority services by the government [31] this context within the pandemic has complicated access barriers to services for women who have been exposed to SV, as the response was directed towards services caring for COVID-19 cases [15, 20].

Regarding the perception of an increase in SV cases that both professionals and young people have, our results show that some young men believe that SV in the family setting in the young population has decreased during lockdown. Some professionals believe that this shows how SV in close settings is not as present in the social imaginary of the young population. In these cases, the existence of violence perpetrated by family members is denied and is combined with the myth that SV only occurs in public places with the most common form of violence, rape [32]. This myth is supported by prejudices and stereotypes of sexual roles and false beliefs that exist around victims and perpetrators, in addition to normalizing men's violence against women [32–34]. According to Waterhouse [35], one such myth, the "real rape" myth, states that most rapes involve a stranger using a weapon attacking a woman violently at night in an isolated, outdoor area, and that women sustain serious injuries from these attacks. By contrast, the majority of reported rape offences (280 cases, 70.7%) were committed by people known to the victim (e.g., domestic and acquaintance rapes), occurred inside a residence, with most victims sustaining no physical injuries from the attack [35].

There is a normalized perception in which the contexts of night life and drinking alcohol facilitate SV [36]. In the case of women, they express this from a position of imagining that situation as victims of sexual abuse. In the case of men, they justify it from a stereotyped place of male impulses as a cause of SV. In fact, they express that SV could have increased after lockdown. Night-life spaces amplify even more this imaginary of flirting and hunting by promoting the sexualization of women's bodies. Therefore, there is a lack of perception of what abuse is, which is normalized in the context of flirting. Similar results are those found by Bennett et al. [16] when reporting that alcohol consumption increases the risk of perpetration and victimization by SV (between 50% and 72% of university sexual assaults).

Finally in our study, a generalized perception has been observed that during the COVID-19 lockdown there has been an increase in SV in a digital setting. Technology has been a fundamental tool to access information, education and work in this time of emergency and it even facilitates access to services for women who are victims of violence, but it may also have been a means used by perpetrators to continue with their behaviour. In this sense, young university students might have been in situations of vulnerability regarding digital SV [16]. In this context, violence facilitated by information and communications technology has spread under the shadow pandemic of violence against women. Women and girls are subject to online violence in the form of physical threats, sexual harassment, stalking, zoom bombing and sex trolling. Traditional stereotypes that continue to exist in social relations between women and men, with sexist values, continue to be projected in online SV.

## Limitations and strengths

The interviewed professionals have only been able to describe what happened from information provided by those who made contact with them, but they cannot provide information on emergent situations of violence that have not reached out to the services they work at. In this sense, the interviewed professionals in this study made no reference to cases of SV against men, probably biased by their own practice so far more focused on female victims. Most young people who were interviewed have provided their perceptions on the impact on SV without having any experience or knowing anyone in this kind of situation. However, their perceptions seem to be in line with what has been observed in other studies. Additionally, we must take into account that the interviews were carried out in the 3 to 6 months after the strict COVID-19 lockdown period in Spain.

This study also takes into account a series of strengths related to applying an emergent design; a theoretical sampling and based on the quality of the information obtained and the degree of saturation finally reached. Furthermore, a description of both the context and the people participating in the interviews is provided. The information has been triangulated by selecting participants with different profiles. In addition, the results are supported by different verbatim citations provided by the heterogeneity of participants in our study. The added value of doing that is by identifying common perceptions despite this heterogeneity, as well as differences between professionals and young people and girls and boys.

## Conclusions

From a perspective of the participating professionals in this study, the lockdown and prevention measures for COVID-19 have a negative impact on attending to and monitoring SV cases. The need to develop protocols for action around violence against women and SV in crisis contexts emerges, as well as the need to improve accessibility to these resources, guaranteeing, for example, their continuity during future. The professionals and young people agree in the fact that there has been a decrease in cases in public spaces due to lockdown and social isolation, although there has been a possible continuity through other means, such as social networks, finding SV in online pornography. Their opinions are different regarding the impact of lockdown within relationships, mainly due to the lack of knowledge of this problem among those who are younger. It is verified that in the imaginary of young men, SV is associated with the public space and the lack of awareness that SV also occurs in the private space and within relationships, where certain sexist SV practices are naturalized. In this sense, investment in public feminist policies and awareness strategies are required in terms of all kinds of SV against women.

## Supporting information

**S1 Table. COREQ checklist.** Consolidated criteria for reporting qualitative research. (DOCX)

**S2 Table. Coding tree.** COVID-19 and sexual violence against women: A qualitative study about young people and professionals' perspectives in Spain. (DOCX)

## Acknowledgments

We want to acknowledge the contributions to this paper made by The CIBER Program of Violence and Young People team is integrated by: Laura Otero-García, Maria José López, Gloria Pérez, Gemma Renart-Vicens, Carmen Saurina, Laura Serra, Laura Vall-Llosera Casanovas.

We also thank study participants, professionals and young people for their contributions. Without their voices and opinions this type of work would not go ahead.

## Author Contributions

**Conceptualization:** Belén Sanz-Barbero, Carmen Vives-Cases.

**Data curation:** Esther Castellanos-Torres.

**Formal analysis:** Esther Castellanos-Torres, Carmen Vives-Cases.

**Funding acquisition:** Belén Sanz-Barbero, Carmen Vives-Cases.

**Investigation:** Esther Castellanos-Torres, Belén Sanz-Barbero, Carmen Vives-Cases.

**Methodology:** Esther Castellanos-Torres, Belén Sanz-Barbero, Carmen Vives-Cases.

**Project administration:** Carmen Vives-Cases.

**Supervision:** Belén Sanz-Barbero, Carmen Vives-Cases.

**Validation:** Esther Castellanos-Torres, Belén Sanz-Barbero, Carmen Vives-Cases.

**Writing – original draft:** Esther Castellanos-Torres, Belén Sanz-Barbero, Carmen Vives-Cases.

**Writing – review & editing:** Esther Castellanos-Torres, Belén Sanz-Barbero, Carmen Vives-Cases.

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
