## [Decision Letter · Decision Letter 0]

7 Oct 2022

PONE-D-22-16766

COVID-19 and sexual violence against women: A qualitative study about young people and professionals’ perspectives in Spain.

PLOS ONE

Dear Dr. Carmen Vives 

Thank you for submitting your manuscript to PLOS ONE. After careful consideration, we feel that it has merit but does not fully meet PLOS ONE’s publication criteria as it currently stands. Therefore, we invite you to submit a revised version of the manuscript that addresses the points raised during the review process.

Even though we have to very different positions regarding the current paper, I agree with reviewer 2 in the sense that there is much work to be done. In its current condition this paper can not be published. I will not reject the paper but provide the authors the opportunity to respond to comments and work in the document. I have other suggestions and comments.

As reviewer 2 says, the qualitative analysis is extremely descriptive and does not provides a theory under the social sciences nor methodological orientation that underpinned the study. Was it guided by grounded theory, discourse analysis, ethnography, phenomenology, or content analysis? This point is fundamental to analyzing and presenting the results of their work.

I suggest that the authors to incorporate elements from the model into the main text; for instance, what experience and training do the researchers enjoy? What information did participants have about the researchers, e.g., their personal goals and reasons for carrying out the study? How many people refused to participate, how many dropped out, and why? What are the main limitations and strenghts of the study?

Having say this, I suggest the authors to work hard on this study and resolve the different issues in order to have a better version of the document.

We look forward to receiving your revised manuscript.

Kind regards,

Cesar Infante Xibille, Ph.D

Academic Editor

PLOS ONE

Journal Requirements:

No authors have competing interests

Additional Editor Comments:

Even though we have to very different positions regarding the current paper I agree with reviewer 2 in the sense that there is much work to be done. In its current condition this paper can not be published. I will not reject the paper but provide the authors the opportunity to respond to comments and work in the document. I have other suggestions and comments.

As reviewer 2 says, the qualitative analysis is extremely descriptive and does not provides a theory under the social sciences nor methodological orientation that underpinned the study. Was it guided by grounded theory, discourse analysis, ethnography, phenomenology, or content analysis? This point is fundamental to analyzing and presenting the results of their work.

I suggest that the authors to incorporate elements from the model into the main text; for instance, what experience and training do the researchers enjoy? What information did participants have about the researchers, e.g., their personal goals and reasons for carrying out the study? How many people refused to participate, how many dropped out, and why? What are the main limitations and strenghts of the study?

Having say this, I suggest the authors to work hard on this study and resolve the different issues in order to have a better version of the document.

Reviewers' comments:

Reviewer's Responses to Questions

**Comments to the Author**

1. Is the manuscript technically sound, and do the data support the conclusions?

Reviewer #1: Yes

Reviewer #2: Partly

2. Has the statistical analysis been performed appropriately and rigorously? 

Reviewer #1: N/A

Reviewer #2: N/A

3. Have the authors made all data underlying the findings in their manuscript fully available?

Reviewer #1: Yes

Reviewer #2: No

4. Is the manuscript presented in an intelligible fashion and written in standard English?

Reviewer #1: Yes

Reviewer #2: No

5. Review Comments to the Author

Reviewer #1: It's a valuable manuscript and reads fluently. I congratulate the authors on the use of this qualitative approach.

The structure of the article is very good and clear.

I have not doubts about the methodology and the discussion.

Reviewer #2: Even though the research theme is quite important and interesting, there are several issues that wil imply a design change of the study and a major revision. I list the following:

1. Number of authors is excessive, there is no way that 10 persons have contribute to write, analyse and discuss information. probably some just did filed work and interviewed participants or transcribed the data.

2. Selection of participants was done through snow ball procedure, which may have skewed the opinions and it is the least rigorous qualitative research design.

3. Refering to results, there are not substancial contributions and several of the findings are obvious, such as that violence in public spaces would be reduced due to confinement and domestic violence or in private spaces would increase, or that digital violence agaisnt women increased,

4. Several of the testimonials are very short and the context is not fully understood.

5. One would think that with 39 interviews and having interviewed young people and professionals there would be much more information and summarizing in 4 categories probably leaves out other relevant information, specially when we read the issues that were explored in the intreviews. This is probably because only one person coded the interview information when it is always better to have more people involved in the coding.

6. Although interviewing both social actors could be a success, there is no integration of what was found because in one case it is about real experiences (those of professionals) and in the other about what the young people imagine.

7. Perhaps it would be better to write an article about what professionals observed and changed in the prevention and care of violence against women and focus not only in the barries but highlight what was learned and what was gained during the pandemic, and another article focused on what young people perceived about the situation, although this seems the weakest information, at least as presented here.

6. PLOS authors have the option to publish the peer review history of their article (what does this mean?). If published, this will include your full peer review and any attached files.

Reviewer #1: **Yes: **Doris Ortega-Altamirano

Reviewer #2: No

---

## [Author Response · Author response to Decision Letter 0]

22 Mar 2023

Authors’ response to reviewers

Reviewer #1: It's a valuable manuscript and reads fluently. I congratulate the authors on the use of this qualitative approach.

The structure of the article is very good and clear.

I have not doubts about the methodology and the discussion.

Thank you for your comment. 

Reviewer #2: Even though the research theme is quite important and interesting, there are several issues that will imply a design change of the study and a major revision. I list the following:

1. Number of authors is excessive, there is no way that 10 persons have contribute to write, analyse and discuss information. probably some just did filed work and interviewed participants or transcribed the data.

This manuscript is part of a large project aimed at analyzing the magnitude, associated factors and perceptions of sexual violence in a sample of young people and stakeholders in different regions of Spain. The authors of this paper make up the research team involved in the project. In the current version of the manuscript, we have revised and improved the information about the authors’ contributions to the papers in the following terms: 

Conceptualization: Carmen Vives-Cases, Esther Castellanos-Torres, Belén Sanz-Barbero contributed altogether to the first ideas of this study and Laura Otero-García, Maria José López, Gloria Pérez, Gemma Renart-Vicens, Carmen Saurina, Laura Serra, and Laura Vall-Llosera Casanovas made relevant contributions to achieve the final formulation of the aim of this study. 

Data Curation: Esther Castellanos-Torres and Carmen Vives-Cases proposed the initial codes and later categories based on the different readings of the transcriptions. Laura Otero-García and Belen Sanz-Barbero helped us in case of discrepancies. 

Formal Analysis: Esther Castellanos-Torres and Laura Otero-García worked on this part with the supervision of Carmen Vives-Cases and Belen Sanz-Barbero. 

Funding Acquisition: Belen Sanz-Barbero, Carmen Vives-Cases, Laura Otero-García, Maria José López, Gloria Pérez, Gemma Renart-Vicens, Carmen Saurina, Laura Serra, Laura Vall-Llosera Casanovas got financial support for the project leading to this publication.

Investigation: Conducting a research and research process, specifically performing the experiments, or data/evidence collection.

Methodology: Belen Sanz-Barbero, Carmen Vives-Cases, Laura Otero-García, Maria José López, Gloria Pérez, Gemma Renart-Vicens, Carmen Saurina, Laura Serra, and Laura Vall-Llosera Casanovas designed the scripts, participant selection criteria and data recruitment strategies. Esther Castellanos-Torres coordinated data collection through personal interviews. 

Project Administration: Carmen Vives-Cases and Belen Sanz-Barbero as PI of the project. 

Resources: Laura Otero-García, Maria José López, Gloria Pérez, Gemma Renart-Vicens, Carmen Saurina, Laura Serra, and Laura Vall-Llosera Casanovas provided contacts with different institutions related to sexual violence and violence against women prevention to identify potential participants.

Software: Not applicable.

Supervision: Carmen Vives-Cases and Belen Sanz-Barbero as PI of the project.

Validation: Not applicable.

Visualization: Carmen Vives-Cases and Belen Sanz-Barbero as PI of the project.

Writing – Original Draft Preparation: Esther Castellanos-Torres, Carmen Vives-Cases and Belen Sanz-Barbero as PI of the project.

Writing – Review & Editing: Laura Otero-García, Maria José López, Gloria Pérez, Gemma Renart-Vicens, Carmen Saurina, Laura Serra, and Laura Vall-Llosera Casanovas contributed to the drafts until they approved the final version. 

2. Selection of participants was done through snow ball procedure, which may have skewed the opinions and it is the least rigorous qualitative research design.

As observed on page 5 of the current version of the manuscript, we provided more information about participants’ recruitment procedure in the following terms: 

The selection criteria for the young people were: 1) age (18-21 years and 22-24 years); 2) sex (women/men); 3) level of studies (university or not); 4) place of origin (migrant and national); and 5) geographical distribution according to region. To select the professionals, the following characteristics of the resources where they worked were considered: 1) type of resource management (public administration or associations); 2) scope of action (approach to SV and youth services); and 3) geographical distribution according to the considered regions (Valencia, the Balearic Islands, Catalonia, Madrid, the Basque Country and Navarre). All these last potential contacts were identified through their institution’s webpages and/or previous contacts of the research team. In the case of young people, through advertisements and flyers distributed in educational environments (high schools, universities, etc.), social media online platforms (Facebook, Instagram, etc.), non-governmental organizations (NGOs) and state institutions working with young people. The criteria to finalize information collection was discourse saturation. Finally, 23 young people and 15 professionals were interviewed.

3. Referring to results, there are not substantial contributions and several of the findings are obvious, such as that violence in public spaces would be reduced due to confinement and domestic violence or in private spaces would increase, or that digital violence against women increased,

The main contribution of this study is putting together professionals’ and young people’s perceptions about the impact of the COVID-19 lockdown in a prevalent issue which is sexual violence against women and men of different sexual orientation. As explained in the introduction of the manuscript (see page 4): 

Despite the valuable contribution made by the testimonies from professionals on the front line in terms of this type of violence, there are still few studies that contrast such professional perceptions with those of women [23]. Qualitative studies have also revealed how COVID-19 preventive measures and the economic crisis hindered and strained intimate relationships among young people, although previous studies focusing on their perspective are also limited [24,25].

The information provided in the paper may contribute to future strategies to cope with this issue in other critical periods as explained with specific implications in the conclusion (page 27): 

From a perspective of participating professionals in this study, the confinement and prevention measures for COVID-19 have a negative impact on attending to and monitoring SV cases. The need to develop protocols for action around violence against women and SV in crisis contexts emerges, as well as the need to improve accessibility to these resources, guaranteeing, for example, their continuity during future. Young people and professionals agree in the fact that there has been a decrease in cases in public spaces due to confinement and social isolation, although there has been a possible continuity through other means, such as social networks, finding SV in online pornography. Their opinions are different regarding the impact of confinement within relationships, mainly due to the lack of knowledge of this problem among those who are younger. It is verified that in the imaginary of young men, SV is associated with the public space and the lack of awareness that SV also occurs in the private space and within relationships, where certain sexist SV practices are naturalized. In this sense, investment in public feminist policies and awareness strategies are required in terms of all kinds of SV against women. 

In addition, in the current version of the manuscript, we have rewritten the results section in order to provide more accurate information according to participant profiles and contexts. 

4. Several of the testimonials are very short and the context is not fully understood.

Thank you very much for this suggestion. The current version of the manuscript provides many of the selected verbatims to improve the information about the context. 

5. One would think that with 39 interviews and having interviewed young people and professionals there would be much more information and summarizing in 4 categories probably leaves out other relevant information, especially when we read the issues that were explored in the interviews. This is probably because only one person coded the interview information when it is always better to have more people involved in the coding.

Data analyses section had been re-written in order to improve the accuracy of the information provided. Please see page 8 the following information: 

A qualitative content analysis was carried out using predefined tree codes that were agreed upon by the research team, based on the study objectives and a preliminary analysis of information from the interviews. After using this initial tree of codes, new codes that emerged were added during the process of analysis process. The tree of codes was adapted during the coding process, by unifying codes according to their discursive similarity and divergence with each informant profile; the emerging topics that were constructed explained the elements that impacted SV during confinement, considering the context from which each informant spoke (professionals and young people). In order to prevent possible bias in coding, we carried out these the following measures were used: double coding of interviews, contrasts of the usefulness of the emerging codes among the analysts and debate sessions with the rest of the members of the research team. The analysis was carried out manually. 

6. Although interviewing both social actors could be a success, there is no integration of what was found because in one case it is about real experiences (those of professionals) and in the other about what the young people imagine.

Thank you for your comment. As we explained in limitations (page 26): Most young people who were interviewed have provided their perceptions on the impact on SV without having any experience or knowing anyone in this kind of situation. However, their perceptions seem to be in line with what has been observed in other studies.

We understand the reviewer’s point, however, we think that it is important to show the different points of view, as they that allow us to compare the perceptions of young people with the experiences of the professionals 

7. Perhaps it would be better to write an article about what professionals observed and changed in the prevention and care of violence against women and focus not only in the barriers but highlight what was learned and what was gained during the pandemic, and another article focused on what young people perceived about the situation, although this seems the weakest information, at least as presented here.

As we explain in the discussion section, the information has been triangulated by selecting participants with different profiles (professionals and young people). In addition, the results are supported by different verbatim citations provided by the heterogeneity of participants in our study. The added value of doing that is by identifying common perceptions despite this heterogeneity, as well as differences between professionals and young people and girls and boys.

---

## [Decision Letter · Decision Letter 1]

16 May 2023

PONE-D-22-16766R1COVID-19 and sexual violence against women: A qualitative study about young people and professionals’ perspectives in Spain.PLOS ONE

Dear Dr. Vives-Cases,

Thank you for submitting your manuscript to PLOS ONE. After careful consideration, we feel that it has merit but does not fully meet PLOS ONE’s publication criteria as it currently stands. Therefore, we invite you to submit a revised version of the manuscript that addresses the points raised during the review process. Please submit your revised manuscript by Jun 30 2023 11:59PM. If you will need more time than this to complete your revisions, please reply to this message or contact the journal office at plosone@plos.org. Please include the following items when submitting your revised manuscript:A rebuttal letter that responds to each point raised by the academic editor and reviewer(s). You should upload this letter as a separate file labeled 'Response to Reviewers'.A marked-up copy of your manuscript that highlights changes made to the original version. You should upload this as a separate file labeled 'Revised Manuscript with Track Changes'.An unmarked version of your revised paper without tracked changes. You should upload this as a separate file labeled 'Manuscript'.

We look forward to receiving your revised manuscript.

Kind regards,

Cesar Infante Xibille, Ph.D

Academic Editor

PLOS ONE

Reviewers' comments:

Reviewer's Responses to Questions

**Comments to the Author**

1. If the authors have adequately addressed your comments raised in a previous round of review and you feel that this manuscript is now acceptable for publication, you may indicate that here to bypass the “Comments to the Author” section, enter your conflict of interest statement in the “Confidential to Editor” section, and submit your "Accept" recommendation.

Reviewer #1: All comments have been addressed

Reviewer #2: All comments have been addressed

Reviewer #3: (No Response)

2. Is the manuscript technically sound, and do the data support the conclusions?

Reviewer #1: Partly

Reviewer #2: Partly

Reviewer #3: Partly

3. Has the statistical analysis been performed appropriately and rigorously? 

Reviewer #1: N/A

Reviewer #2: N/A

Reviewer #3: N/A

4. Have the authors made all data underlying the findings in their manuscript fully available?

Reviewer #1: Yes

Reviewer #2: No

Reviewer #3: (No Response)

5. Is the manuscript presented in an intelligible fashion and written in standard English?

Reviewer #1: Yes

Reviewer #2: No

Reviewer #3: (No Response)

6. Review Comments to the Author

Reviewer #1: It is an article that provides vulnerable overpopulation information, but not affected by sexual violence at the time of the interview. However, the authors must demonstrate that their study has sufficient scientific rigor and practical value to be replicated.

There is discrepancy in the number of professionals interviewed in the summary 16 are mentioned on page 5, line 98 says 15. Table 2 shows 16 and is understood to be 15 due to an interview done in two parts.

There is a discrepancy in the abstract which indicates 39 participants and on page 7, line 120 indicates 12 video calls and 26 interviews, giving a total of 38.

Two topics of interviews between professionals and young people are consistent, three not. Professionals theme 1 - young people theme 3; Professionals theme 3 - young people theme 4. However, there is no consistency with the display of results presenting for young people themes 1 and 2: perceptions about violence and experience (indirect) about the problem. Authors should elaborate on explaining the selection of topics included in the analysis (triangulation).

The method should incorporate information on the three code trees and add a figure with these trees to understand the methodological and theoretical meaning of the coding made by the authors.

Transcription and encoding validation should be reported. Validation is referred to as "not applicable".

Page 8, line 145 says "prevent bias" what kind of bias do you mean? It is recommended to describe the profile of the researchers who did the coding, planned the code trees and did the coding. This would strengthen the analysis and processing of data.

The method should incorporate information on the three code trees and add a figure with these trees to understand the methodological and theoretical meaning of the coding made by the authors.

Explain whether or not the coding trees were used for the presentation of results. Only two groups of results are shown. The first refers to the "Impact of response of health professionals". This impact refers to your performance during confinement or the answers given during the interview. Result concerning "suspension of service" gives appearance of impact on the service, however the title of the paragraph does not indicate it. . And the result relative to "other expressions of sexual violence" refer to the time after confinement (after), gives appearance of impact response of professionals in the interview. Please clarify the meaning of the first paragraph of results on page 8.

Reviewer #2: Even though all comments have been answered there are still some minor issues to be considered before publication.

1) Number of authors (10) is still excesive and even if they have explained each author contribution, not all of their contributions imply that they mignt be considered as authors. Only seven of the ten authors are mentioned as they do write and review the manuscript, and first and corresponding authors are not included there. The suggestion is to ruduce number of authors.

2) I suggest to list final codes that ere used for the analysis in the Data analysis section starting on line 137

3) I suggest to delay those testimonies of just one line of text, because they don´t provide sustancial information, besides on line 164, what are the purple spots? Please check the manuscript, that´s quite long now, and take away all those so brief testimonies.

4) Be careful because manuscript has too many testimonies, select just one or two in each paragraph were you decide to include them. This is a risky practice in qualitative manuscripts: having an excess of testimonies and little analysis.

5) Congratulations for including a section of Opportunities of online services

6) Results refering to young population are still weak, try to relate them with professionals experiences.

Reviewer #3: The objective of the study requires greater precision. It is not clear how this work represents the reality in Spain since it is a qualitative work.

It talks about the responsibilities of care of sexual violence as well as the consequences but does not delve into these issues in the introduction.

It seems to be a work on care processes and how they have been impacted by the covid pandemic. I see it as a work on the mechanisms that have facilitated and hindered attention and that is where it should focus.

There is a lack of definition in the conceptual and reference framework of the work. A definition of sexual violence is provided, but apparently that is not the focus of the article, but rather the impact of the pandemic on attention to sexual violence.

This lack of definition is reflected throughout the document, since by not presenting the conceptual framework and the research question that guides the investigation, multiple topics are presented with little integration in the development of the work.

There is also no adequate justification for why it is sought to compare the perceptions of users and service providers. Finally, this comparison is not achieved in the development of the work.

The lack of clarity in the conceptual framework makes us wonder where the dimensions explored in the interviews come from and how they are defined. In this same sense, how is it that they manage to compare the experiences of different informants if the same dimensions are not explored in the interviews?

There are too many extremely concise testimonies, of no more than one line. It is suggested to eliminate them. Having so many testimonies reduce the possibility of reflection and the opportunity to go further in the interpretation of the testimonies is lost.

It will also be important to present information that allows us to know if the informant is a man, a woman, age, location, among others. Numeric identifiers do not contribute much in a qualitative work

Why are the results presented in relation to different topics that apparently were not explored in the interviews? at least they are not explicit in the text. It is said that codes arose.. what were they?

There is no proper integration of the topics throughout the development of the study. I think that the lack of clarity by not having a research question and not presenting a frame of reference explains why many topics are presented and their integration is not clear.

It is difficult to identify the main argument and contribution of the work. From my point of view there is no comparison of the perspectives or experiences between users of services and providers. It is also not possible to delve into a central element since multiple topics are presented. For example. There is talk of myths, prejudices and stereotypes, but these issues are not considered as relevant in the study, nor is it identified where they arise from. If they were substantive elements, they should have been placed in their correct dimension and the text should be developed on this.

In order to consider the work for publication, authors must clearly define three things: frame of reference, question, and research objective. This will allow them to reorder the work so that it is much more solid and there is a central argument in its development.

7. PLOS authors have the option to publish the peer review history of their article (what does this mean?). If published, this will include your full peer review and any attached files.

Reviewer #1: **Yes: **Doris V. Ortega-Altamirano

Reviewer #2: No

Reviewer #3: No

---

## [Author Response · Author response to Decision Letter 1]

28 Jun 2023

Reviewer #1: It is an article that provides vulnerable overpopulation information, but not affected by sexual violence at the time of the interview. However, the authors must demonstrate that their study has sufficient scientific rigor and practical value to be replicated.

Thank you for your valuable comments. We are sure the manuscript have improved thanks to them. 

There is discrepancy in the number of professionals interviewed in the summary 16 are mentioned on page 5, line 98 says 15. Table 2 shows 16 and is understood to be 15 due to an interview done in two parts.

You are right. We corrected the mistake in both the abstract and the table. 

There is a discrepancy in the abstract which indicates 39 participants and on page 7, line 120 indicates 12 video calls and 26 interviews, giving a total of 38.

You are right. We corrected the mistake which was related with the previous comment.

Two topics of interviews between professionals and young people are consistent, three not. Professionals theme 1 - young people theme 3; Professionals theme 3 - young people theme 4. However, there is no consistency with the display of results presenting for young people themes 1 and 2: perceptions about violence and experience (indirect) about the problem. Authors should elaborate on explaining the selection of topics included in the analysis (triangulation).

It was an error that we committed summarizing the content of the interview guides. We rewrote this section in order to make clear the consistency between them. In the current version of the manuscript, it is described in the following terms (see page 7): 

Two thematic scripts were adapted, one aimed at the young population and the other at professionals. The scripts were tested and adapted in the early stages of the fieldwork. The final structure of the script aimed at professionals focused on the perceptions of the informants on the following dimensions: 1) description and assessment of the resource from which where they work; 2) perception on sexual violence and young people’s victimization; 3) young people’s difficulties and facilitators in accessing and using the resources by young people; and 4) the impact perceived of SV throughout the COVID-19 pandemic in SV among young people and their own services and work. The final structure of the script aimed at young people focused on the perceptions of the informants on the following dimensions: 1) perceptions on sexual violence; 2) experience with this problem; 3) knowledge and assessment (self-perceived difficulties and facilitators) of the existing related resources; and 4) the impact perceived of SV throughout the COVID-19 pandemic in SV among young people and the existing SV-related services. This study is focused on describing and comparing the professionals’ and young women and men’s responses to the fourth topic of both interview guides. 

The method should incorporate information on the three code trees and add a figure with these trees to understand the methodological and theoretical meaning of the coding made by the authors.

We included the tree code in the current version of the manuscript (See S2. Table. Tree Code). 

Transcription and encoding validation should be reported. Validation is referred to as "not applicable".

You are right. We added in this section of the table about authors’ contribution the following measures related with validation: The following measures were used: double coding of interviews, contrasting the usefulness of the emerging codes with CVC and BSB in the analyst and debate sessions with the rest of the members of the research team to clarify discrepancies between the main analysts (the coding tree is shown as a table in [Supplementary-material pone.0289402.s002]. Coding tree.docx) (see authors’ contribution in the response to reviewer 2). 

Page 8, line 145 says "prevent bias" what kind of bias do you mean? 

We wanted to refer measures to assure the reliability. In the current version of the manuscript, we eliminated this concept of “bias”. 

It is recommended to describe the profile of the researchers who did the coding, planned the code trees and did the coding. This would strengthen the analysis and processing of data. The method should incorporate information on the three code trees and add a figure with these trees to understand the methodological and theoretical meaning of the coding made by the authors.

Thank you for your comment. We added all this information in the current version of the manuscript. On page 8, you will find the following information: 

A qualitative content analysis was carried out by two of the authors, ECT and CVC (both sociologists and experts in gender violence, sexual violence and qualitative research), using a predefined tree of codes that were agreed upon by all authors (experts in gender violence and sexual violence) and based on the study objectives and a preliminary analysis of information from the interviews. After using this initial tree of codes, new codes that emerged were added during the analysis process. The tree of codes was adapted during the coding process, by unifying codes according to their discursive similarity and divergence with each informant profile; the emerging topics that were constructed explained the elements that impacted SV during lockdown, considering the context from which each informant spoke (professionals and young people). The following measures were used: double coding of interviews, contrasting the usefulness of the emerging codes with CVC and BSB in the analyst and debate sessions with the rest of the members of the research team to clarify discrepancies between the main analysts (the coding tree is shown as a table in [Supplementary-material pone.0289402.s002]. Coding tree.docx). The analysis was carried out manually. 

Explain whether or not the coding trees were used for the presentation of results.

Yes, the results section is guided by the codes and categories included in the coding tree which had been added in the current version of the manuscript ([Supplementary-material pone.0289402.s002]. Tree Code). As it is described in results, two main themes emerged: the impact of professional answers and the impact perceived by young people and professionals regarding SV. These themes are described in results section and included in the coding tree. In each theme, different categories were integrated (also present in the Tree Code. We rewrote the first paragraph of result to make this clearer (see page 9) in the following terms: 

Two main themes have been identified throughout this study, first of all the impact COVID-19 on professional answers; and second of all, the impact regarding SV. The first one was mainly referred by the professionals, while the second one referred to both the professionals and young people’s perceptions. The first theme has been categorized into two aspects related to the negative and positive effects of the COVID-19 preventive measures. These two aspects were described in the following sections: 1) from paralyzing direct activity due to lockdown to searching for alternatives for support and intervention in SV; 2) opportunities of online services: the positive aspects of online interventions; and 3) victim-related challenges. The second theme includes agreements and disagreements among professionals and young women and men due to their knowledge or misunderstandings about SV, intimate partner sexual violence, other types of SV, its visibility (or invisibility) and the settings where cases occurred during and after the COVID-19 lockdown. The following categories are described in the results: 1) SV inside and outside the family setting; 2) an increase in SV in digital settings; and 3) lockdown and a decrease in SV in public; and 4) SV cases silenced during the COVID-19 lockdown. 

Only two groups of results are shown. 

They are the two main themes identified after assembled codes and categories. 

The first refers to the "Impact of response of health professionals". This impact refers to your performance during confinement or the answers given during the interview. Result concerning "suspension of service" gives appearance of impact on the service, however the title of the paragraph does not indicate it. 

We did not include “suspension of service” in the title because the activities of these professionals continued as they were considered an “essential” sector. It is true that part of their programmed were cancelled due to the prohibition of doing presential and direct contact activities, but they were transformed in order to continue with their contact with the young population. We think we should keep the original title of “From paralyzing direct activity due to confinement lockdown to searching for alternatives for support and intervention in SV.” to make clear that their activity was not suspended but suffered important changes. 

And the result relative to "other expressions of sexual violence" refer to the time after confinement (after), gives appearance of impact response of professionals in the interview. Please clarify the meaning of the first paragraph of results on page 8.

We agree with you that it is confusing, and we rewrote the title of the category. In the current version of the manuscript, these results are part of the subsection entitled: “Victims-related challenges”. 

Reviewer #2: Even though all comments have been answered there are still some minor issues to be considered before publication.

1) Number of authors (10) is still excesive and even if they have explained each author contribution, not all of their contributions imply that they mignt be considered as authors. Only seven of the ten authors are mentioned as they do write and review the manuscript, and first and corresponding authors are not included there. The suggestion is to ruduce number of authors.

As we explained before, this manuscript is part of a large project, and the authors gave to it and the manuscript the needed experience and multidisciplinary. We understand your concerns about the authorship, but we can’t change it in the second revision of the manuscript. We reviewed the information about the authors’ contributions in order to make them clearer: 

Conceptualization Carmen Vives-Cases, Esther Castellanos-Torres, Belén Sanz-Barbero contributed altogether to the first ideas of this study and Laura Otero-García, Maria José López, Gloria Pérez, Gemma Renart-Vicens, Carmen Saurina, Laura Serra, and Laura Vall-Llosera Casanovas made relevant contributions to achieve the final formulation of the aim of this study. 

Data Curation Esther Castellanos-Torres and Carmen Vives-Cases proposed the initial codes and later categories based on the different readings of the transcriptions. Laura Otero-García and Belen Sanz-Barbero helped us in case of discrepancies. 

Formal Analysis Esther Castellanos-Torres worked on this part with the supervision of Carmen Vives-Cases and Belen Sanz-Barbero. 

Funding Acquisition Belen Sanz-Barbero, Carmen Vives-Cases, Laura Otero-García, Maria José López, Gloria Pérez, Gemma Renart-Vicens, Carmen Saurina, Laura Serra, Laura Vall-Llosera Casanovas got financial support for the project leading to this publication.

Investigation The data/evidence collection was conducted by two interviewers by the supervision of Carmen Vives-Cases and Esther Castellanos-Torres. 

Methodology Belen Sanz-Barbero, Carmen Vives-Cases, Laura Otero-García, Maria José López, Gloria Pérez, Gemma Renart-Vicens, Carmen Saurina, Laura Serra, and Laura Vall-Llosera Casanovas designed the scripts, participant selection criteria and data recruitment strategies. Esther Castellanos-Torres coordinated data collection through personal interviews. 

Project Administration Carmen Vives-Cases and Belen Sanz-Barbero as PI of the project. 

Resources Laura Otero-García, Maria José López, Gloria Pérez, Gemma Renart-Vicens, Carmen Saurina, Laura Serra, and Laura Vall-Llosera Casanovas provided contacts with different institutions related to sexual violence and violence against women prevention to identify potential participants.

Software The analyses was manually performed. 

Supervision Carmen Vives-Cases and Belen Sanz-Barbero as PI of the project.

Validation Double coding of interviews, contrasts of the usefulness of the emerging codes among the analyst and debate sessions with the rest of the members of the research team to clarified discrepancies between the main analysts

Visualization Carmen Vives-Cases and Belen Sanz-Barbero as PI of the project.

Writing – Original Draft Preparation Esther Castellanos-Torres, Carmen Vives-Cases and Belen Sanz-Barbero as PI of the project.

Writing – Review & Editing Esther Castellanos-Torres, Carmen Vives-Cases, Belen Sanz-Barbero, Laura Otero-García, Maria José López, Gloria Pérez, Gemma Renart-Vicens, Carmen Saurina, Laura Serra, and Laura Vall-Llosera Casanovas contributed to the drafts until they approved the final version. 

2) I suggest to list final codes that were used for the analysis in the Data analysis section starting on line 137.

Thank you for your suggestion. We added the Tree Code in [Supplementary-material pone.0289402.s002]. Coding tree. docx. 

3) I suggest to delay those testimonies of just one line of text, because they don´t provide sustancial information, besides on line 164, what are the purple spots? Please check the manuscript, that´s quite long now, and take away all those so brief testimonies.

Ok. We eliminated all quotations which were not longer than one line in the current version of the manuscript. 

4) Be careful because manuscript has too many testimonies, select just one or two in each paragraph were you decide to include them. This is a risky practice in qualitative manuscripts: having an excess of testimonies and little analysis.

We agree with you but we added them due to a suggestion received in the first revision. In the current version of the manuscript, we did the recommended selection of quotations. 

5) Congratulations for including a section of Opportunities of online services

Thank you very much. 

6) Results refering to young population are still weak, try to relate them with professionals experiences.

As you can see in the current version of the manuscript, all themes are described integrating perceptions of professionals, young women and young men. 

Reviewer #3: The objective of the study requires greater precision. It is not clear how this work represents the reality in Spain since it is a qualitative work.

It talks about the responsibilities of care of sexual violence as well as the consequences but does not delve into these issues in the introduction.

We rewrote the objective in the abstract and the introduction in order to increase its accuracy (See pages 2 and 4): we aim to analyse SV in the COVID-19 lockdown among young people and SV-related services from the perspective of professionals and young people from different sectors in Spain with responsibilities in attending SV and other forms of violence against women-related.

It seems to be a work on care processes and how they have been impacted by the covid pandemic. I see it as a work on the mechanisms that have facilitated and hindered attention and that is where it should focus.

There is a lack of definition in the conceptual and reference framework of the work. A definition of sexual violence is provided, but apparently that is not the focus of the article, but rather the impact of the pandemic on attention to sexual violence. This lack of definition is reflected throughout the document, since by not presenting the conceptual framework and the research question that guides the investigation, multiple topics are presented with little integration in the development of the work.

We rewrote the introduction section in order to improve the focus of the manuscript and justify why sexual violence is approached, why in young people and the relevance of COVID-19 lockdown impact in this problem and SV-related services. 

There is also no adequate justification for why it is sought to compare the perceptions of users and service providers. Finally, this comparison is not achieved in the development of the work.

We added an additional justification after describing the study objective (see on page 4): Studying the perceptions and imaginaries of both professionals and young women and men may contribute to the limited research about SV among young people and the COVID-19 impact on SV that include both perspectives.

As you can see in the current version of the manuscript, the second main theme of results is described integrating perceptions of professionals, young women and young men in order to make the comparison more evident (see pages 14-20).

The lack of clarity in the conceptual framework makes us wonder where the dimensions explored in the interviews come from and how they are defined. In this same sense, how is it that they manage to compare the experiences of different informants if the same dimensions are not explored in the interviews?

Most of the compared dimensions were included in the interviews. However, It was an error that we committed summarizing the content of the interview guides. We rewrote this section in order to make clear the consistency between them. In the current version of the manuscript, it is described in the following terms (see page 7): 

Two thematic scripts were adapted, one aimed at the young population and the other at professionals. The scripts were tested and adapted in the early stages of the fieldwork. The final structure of the script aimed at professionals focused on the perceptions of the informants on the following dimensions: 1) description and assessment of the resource where they work; 2) perception on sexual violence and young people’s victimization; 3) young people’s difficulties and facilitators in accessing and using resources; and 4) the impact perceived of SV throughout the COVID-19 pandemic among young people and their own services and work. The final structure of the script aimed at young people focused on the perceptions of the informants on the following dimensions: 1) perceptions on sexual violence; 2) experience with this problem; 3) knowledge and assessment (self-perceived difficulties and facilitators) of the existing resources; and 4) the impact perceived of SV throughout the COVID-19 pandemic among young people and the existing SV-related services. This study is focused on describing and comparing the professionals’ and young women and men’s responses to the fourth topic of both interview guides. 

There are too many extremely concise testimonies, of no more than one line. It is suggested to eliminate them. Having so many testimonies reduce the possibility of reflection and the opportunity to go further in the interpretation of the testimonies is lost.

We eliminated all quotations which were not longer than one line in the current version of the manuscript and did a selection of the most relevant ones. 

It will also be important to present information that allows us to know if the informant is a man, a woman, age, location, among others. Numeric identifiers do not contribute much in a qualitative work

We change the labels in order to identified the sex of the informant in the quotations. The rest of details are included in table 1 and 2 and we decided not including them in the labels of the quotations to make them shorter. 

Why are the results presented in relation to different topics that apparently were not explored in the interviews? at least they are not explicit in the text. It is said that codes arose.. what were they?

We improved the interview guide description and the analyses section in order to clarify them. We also added the Tree Code in “[Supplementary-material pone.0289402.s002]. Coding tree. docx”. In the current version of the manuscript, you can read: 

(Page 7): Two thematic scripts were adapted, one aimed at the young population and the other at professionals. The scripts were tested and adapted in the early stages of the fieldwork. The final structure of the script aimed at professionals focused on the perceptions of the informants on the following dimensions: 1) description and assessment of the resource where they work; 2) perception on sexual violence and young people’s victimization; 3) young people’s difficulties and facilitators in accessing and using resources; and 4) the impact perceived of SV throughout the COVID-19 pandemic among young people and their own services and work. The final structure of the script aimed at young people focused on the perceptions of the informants on the following dimensions: 1) perceptions on sexual violence; 2) experience with this problem; 3) knowledge and assessment (self-perceived difficulties and facilitators) of the existing resources; and 4) the impact perceived of SV throughout the COVID-19 pandemic among young people and the existing SV-related services. This study is focused on describing and comparing the professionals’ and young women and men’s responses to the fourth topic of both interview guides.. 

Page 8: A qualitative content analysis was carried out by two of the authors, ECT and CVC (both sociologists and experts in gender violence, sexual violence and qualitative research), using a predefined tree of codes that were agreed upon by all authors (experts in gender violence and sexual violence) and based on the study objectives and a preliminary analysis of information from the interviews. After using this initial tree of codes, new codes that emerged were added during the analysis process. The tree of codes was adapted during the coding process, by unifying codes according to their discursive similarity and divergence with each informant profile; the emerging topics that were constructed explained the elements that impacted SV during lockdown, considering the context from which each informant spoke (professionals and young people). The following measures were used: double coding of interviews, contrasting the usefulness of the emerging codes with CVC and BSB in the analyst and debate sessions with the rest of the members of the research team to clarify discrepancies between the main analysts (the coding tree is shown as a table in [Supplementary-material pone.0289402.s002]. Coding tree.docx). The analysis was carried out manually. 

There is no proper integration of the topics throughout the development of the study. I think that the lack of clarity by not having a research question and not presenting a frame of reference explains why many topics are presented and their integration is not clear.

The aim of this study was to analyze the perceptions and imaginaries that young people and professionals of different sectors with responsibilities in attending violence-related resources in Spain have about the main consequences of COVID-19 lockdown on SV in Spain. We considered that results related professionals’ perception about confinement impact in their own activities, programs and services and their experiences leading with compulsory changes in their practices and the comparison of their opinions and the young people ones about the consequences on sexual violences, its different forms (visible and invisible) and the accessibility to SV-related services (barriers and facilitators) answer this objective. This study is in line with others which have been already performed (for example references from 14 to 30 cited in the section of bibliography). As we said before, we improved the introduction in order to clarify the research question. 

It is difficult to identify the main argument and contribution of the work. From my point of view there is no comparison of the perspectives or experiences between users of services and providers. 

As we mentioned before, we added an additional justification after describing the study objective (page 4). 

It is also not possible to delve into a central element since multiple topics are presented. For example. There is talk of myths, prejudices and stereotypes, but these issues are not considered as relevant in the study, nor is it identified where they arise from. If they were substantive elements, they should have been placed in their correct dimension and the text should be developed on this.

They are relevant in relation with the topic as it is explained in the introduction, but not substantive elements according to our study aim. 

In order to consider the work for publication, authors must clearly define three things: frame of reference, question, and research objective. This will allow them to reorder the work so that it is much more solid and there is a central argument in its development.

We tried to attend this suggestion and rewrote different sections of the manuscript in order to clarify these three important issues.

---

## [Editor Report · Decision Letter 2]

12 Jul 2023

PONE-D-22-16766R2COVID-19 and sexual violence against women: A qualitative study about young people and professionals’ perspectives in Spain.PLOS ONE

Dear Dr. Carmen Vives

Thank you for submitting your manuscript to PLOS ONE. After careful consideration, we feel that it has merit but does not fully meet PLOS ONE’s publication criteria as it currently stands. Therefore, we invite you to submit a revised version of the manuscript that addresses the points raised during the review process.

My only concern is the number of authors in this paper. For a paper like this I should only consider a maximum of six authors. The justification the authors provide to justify the excessive number of authors is not sufficient. Please include no more than six authors.

We look forward to receiving your revised manuscript.

Kind regards,

Cesar Infante Xibille, Ph.D

Academic Editor

PLOS ONE

---

## [Author Response · Author response to Decision Letter 2]

13 Jul 2023

We decided to keep in the authorship the main authors (the first, the second and the last and corresponding one) and integrated the rest in acknowledge as it is specified in the authors’s guidelines for group authorships.

---

## [Editor Report · Decision Letter 3]

19 Jul 2023

COVID-19 and sexual violence against women: A qualitative study about young people and professionals’ perspectives in Spain.

PONE-D-22-16766R3

Dear Dr. Carmen Vives-Cases,

We’re pleased to inform you that your manuscript has been judged scientifically suitable for publication and will be formally accepted for publication once it meets all outstanding technical requirements.

Kind regards,

Cesar Infante Xibille, Ph.D

Academic Editor

PLOS ONE

---

## [Editor Report · Acceptance letter]

24 Jul 2023

PONE-D-22-16766R3 

COVID-19 and sexual violence against women: A qualitative study about young people and professionals’ perspectives in Spain. 

Dear Dr. Vives-Cases:

I'm pleased to inform you that your manuscript has been deemed suitable for publication in PLOS ONE. Congratulations! Your manuscript is now with our production department. 

Kind regards, 

on behalf of

Dr. Cesar Infante Xibille 

Academic Editor

PLOS ONE